# Predicting the Printability in Selective Laser Melting with a Supervised Machine Learning Method

**DOI:** 10.3390/ma13225063

**Published:** 2020-11-10

**Authors:** Yingyan Chen, Hongze Wang, Yi Wu, Haowei Wang

**Affiliations:** 1State Key Laboratory of Metal Matrix Composites, Shanghai Jiao Tong University, No. 800 Dongchuan Road, Shanghai 200240, China; cyy1996@sjtu.edu.cn (Y.C.); hwwang@sjtu.edu.cn (H.W.); 2School of Materials Science and Engineering, Shanghai Jiao Tong University, No. 800 Dongchuan Road, Shanghai 200240, China

**Keywords:** selective laser melting, machine learning, defect detection, printability prediction

## Abstract

Though selective laser melting (SLM) has a rapidly increasing market these years, the quality of the SLM-fabricated part is extremely dependent on the process parameters. However, the current metallographic examination method to find the parameter window is time-consuming and involves subjective assessments of the experimenters. Here, we proposed a supervised machine learning (ML) method to detect the track defect and predict the printability of material in SLM intelligently. The printed tracks were classified into five types based on the measured surface morphologies and characteristics. The classification results were used as the target output of the ML model. Four indicators had been calculated to evaluate the quality of the tracks quantitatively, serving as input variables of the model. The data-driven model can determine the defect-free process parameter combination, which significantly improves the efficiency in searching the process parameter window and has great potential for the application in the unmanned factory in the future.

## 1. Introduction

Selective laser melting (SLM), a powder-based additive manufacturing (AM) which is capable of fabricating components with complex shape, has now been increasingly used to produce metallic parts especially in the aerospace field [1]. Besides the material, achieving defect-free AM part is also dependent on the process parameters. Inappropriate parameters may result in defects in the track, e.g., discontinuity [2], balling [3,4], humping [5], pores [6], and spatters [7], which will lead to a decrease in performance of the SLM-fabricated part. Therefore, figuring out suitable process parameters for a defect-free track is the firm foundation of a SLM-fabricated part with excellent performance.

For the single track, laser power and scan speed during the process is of great importance. Various efforts have been made to obtain the suitable combinations of them. Yadroitsev et al. [8,9] obtained a parameter window for stable single track in SLM of stainless steel by experiment and the subsequent surface morphology analysis. The results showed that the broken single vectors were formed at a low scan speed, while balling phenomenon occurred at a higher one. In a similar approach, Gu et al. [10] obtained a parameter window for four types of tracks for direct metal laser sintering of 316L SS powder. They found out that large laser power and low scan speed led to stable melt pool and continuous track. Further, a quantitative analysis method based on melt pool dimension was used to obtain the appropriate processing parameters. Scime et al. [11] studied the morphology and size distribution of the melt pool and figured out a classification method of combination quantitative analysis with qualitative analysis to judge whether the parameter combination was suitable for SLM process. Johnson et al. [12] set up a finite element thermal model to calculate the melt pool dimension, and used the quantitative data to detect the defect formation and predict the printability of material, where the theoretical results agreed well with the experimental ones. As the developments of the modern manufacturing industry, there is an increasing demand for automation, stability, and repeatability [13,14]. Intelligently identifying the track defect and figuring out the suitable parameter combinations are of high demand. Most of these previous works adopted a method involving subjective assessment of the experimenters and were not suitable for unmanned factory in the future.

Machine learning (ML), a data-driven method with the superiority of learning from the given data and making the decision automatically by itself [15,16], is capable to decrease human intervention and carry out the prediction automatically in modern manufacturing [17], which may solve the challenge mentioned above. Wu et al. [18] presented an approach to predict the surface roughness in fused deposition modeling by combination real-time monitoring with random forest network (RFN) ML model. Kappes et al. [19] also used the RFN ML model to study the relationship between the powder bed fusion (PBF) process and the porosity of the fabricated part. Baturynska et al. [20] proposed a conceptual framework to optimize process parameters by a method of combining ML with finite element model. Gobert et al. [21] described an in-situ defect detection and process monitoring strategy for PBF using a method of combining ML with high-resolution imaging. ML model has also been used in the research of the other AM technologies [22,23,24,25,26,27] and friction stir welding [28,29] in recent years, which validates the potential of ML for better manufacturing process.

Here, we propose a supervised machine learning (ML) method to detect the track defect and predict the printability of material in SLM intelligently, which can figure out the suitable parameter combinations to fabricate the defect-free track and therefore the part with excellent performance. Thirty-three single tracks under different parameter combinations of laser power and scan speed were fabricated. Then the formed tracks were classified according to their surface morphologies and characteristics. Based on the analysis of the surface morphology, four indicators had been calculated to evaluate the quality of the tracks quantitatively. A backpropagation-based neural network model, one of the ML methods, was then developed. The classification results were used as target output while the four indicators serve as input variables. Finally, defect detection and printability prediction can be realized by using the trained neural network. Taking advantage of the automatic feature-learning capacity of machine learning, this approach has great potential to determine the printable parameter window intelligently in the unmanned factory.

## 2. Materials and Methods

### 2.1. The General Workflow of the Prediction Method

Figure 1 shows the schematic of the prediction method. Machine learning (ML) was used in this prediction method. ML trains the model with input data and uses the trained model to predict the new input data. Before the prediction process, usually there will be a testing process to check the accuracy of the trained model. Based on the ML model and the combination quantitative analysis with qualitative analysis, the parameter window of single track can be intelligently predicted.

The workflow started from the SLM experiments and classification of the formed tracks. The printed tracks are classified into five types based on the measured surface morphologies and characteristics. The classification results are used as the target output of the ML model. Four indicators have been calculated to evaluate the quality of the tracks quantitatively, serving as input variables of the model. Specifically, a backpropagation-based neural network model was set up for the prediction in this work. The newly developed prediction method consists of three steps:Extracting data from the experimental results: The formed tracks are sorted according to their surface quality, followed by calculating the value of the four evaluation indicators. Evaluation indicators and classification results are used as input variables and target output, respectively, of the following neural network model.Training the ML model: The samples are randomly selected for the training and testing process. A backpropagation-based neural network model was set up for the prediction task in this work. The factors which could affect the prediction accuracy includes the size of the database, the learning algorithm, and the network structure.Practical prediction to guide the SLM process: After inputting the related evaluation indicators of the parameters to be examined, a value representing the track’s possibility of having defects will be returned, which could help to guide the SLM process.

### 2.2. SLM Processing

The homemade in-situ 6 wt.% nano-TiB2 reinforced AlSi10Mg composite was used in this study. The as-cast composite was synthesized by an two-step in-situ mixed salt reaction method which were preparing TiB_2_/Al composites through the exothermic reaction via mixture salts of K_2_TiF_6_ and KBF_4_ and obtaining as-cast ingots by adding high purity ingots of alloying elements [30]. Then the powder was produced by vacuum gas atomization and screened. The powder used in the SLM experiment has a generally spherical shape and a size ranging from 22.55 μm to 49.98 μm. The SLM machine (ProX^®^DMP200, 3D Systems, Inc., Rock Hill, SC, USA) employed in this research was equipped with a fiber laser with a wavelength of 1070 nm and a spot diameter of 75 μm. Before the printing experiment, pure argon gas was delivered into the chamber until the oxygen content detected by the built-in oxygen sensor of the SLM machine reached the range between 0 and 1 ppm, aiming to mitigate the oxidation content in melt pool during the fabrication process. A powder layer thickness of 30 μm was set and the continuous laser mode was used during the fabrication. Then thirty-three parameter sets were used to produce single tracks whose length were all 1 cm on a bare plate of 5083 Al alloy (nominally Al-4.6%Mg). The effect of scan speed (V) in the range of 200 to 2200 mm/s was investigated, and three laser powers (P) (90 W, 195 W, and 300 W) were adopted.

### 2.3. Characterization of the Tracks

The surface morphologies of the thirty-three single tracks were obtained through Laser Scanning Confocal Microscope (Zeiss LSM 900, Carl Zeiss AG, Jena, Germany). Simultaneously, for each single track, a 3D matrix file with the dimensions of X, Y, and Z, respectively, containing the microscope information, was also exported. X and Y referred to the directions in the measurement plane while Z referred to the relative height to the standard plate. We used the MATLAB software (MATLAB R2020b, The MathWorks, Natick, MA, USA) to process the data, providing information about the outline of the track’s cross-sections perpendicular to the laser beam travel direction. The number of cross-sections extracted from a track was related to the enlargement factor and lateral resolution of the measurement equipment during the scanning process. In this work, 2048 cross-sections were extracted from a track.

Subsequently data of the width and height of each cross-section were extracted for each track. The average and standard deviation values of both width and height were then calculated and used to obtain the ratios for each track, $R_{w}=s_{w}/\bar{w}$, $R_{h}=s_{h}/\bar{h}$, where $\bar{w}$ and $\bar{h}$ corresponded to the average width and height while $s_{w}$ and $s_{h}$ corresponded to the standard deviation of the width and height. The formulas for calculating the average and standard deviation values were as follows [31]:(1)$$\bar{x}=\frac{\sum_{i=1}^{n}x_{i}}{n}$$
(2)$$s=\sqrt{\frac{\sum_{i=1}^{n}(x_{i}-\bar{x}{)}^{2}}{n-1}}$$
where $x_{i}$ was the width or height of each cross-section, n was the total number of the cross-section, $\bar{x}$ was the average width ($\bar{w}$) or height ($\bar{h}$), s was the standard deviation of the width ($s_{w}$) or the height ($s_{h}$).

### 2.4. Neural Network Model

Among all the ML models, neural network (NN) was selected in this work. The neural network is composed of many interconnected nodes which are arranged in layers (input layer, hidden layers, output layer), and a transfer function is used to introduce nonlinear relationship between the layers. The nodes receive input from an external source or nodes in the previous layer and produce an output with the transfer function. Each connection between two adjacent layers has an associated weight [32] to represent its relative importance to other connections. Additionally, there is a bias term with a trainable value attached to the nodes of hidden layers and output layer and serves as an intercept term [33]. The training of the neural network model is essentially an iterative process of adjusting both the connection weights and bias terms to fit the transfer function and minimize a predetermined loss function.

For the training process, we used a backpropagation algorithm called gradient descent with momentum (GDM). GDM algorithm is an optimization of the standard gradient descent approach, which considers the gradient of both the current and the previous step when updating the weights and bias terms [34,35]. The values of weights and bias terms were updated with the following equations [36]:(3)$$m_{0}=0$$
(4)$$m_{k+1}=\gamma m_{k}+\eta \times \frac{\partial E}{\partial w_{k}}$$
(5)$$w_{k+1}=w_{k}-m_{k+1}$$
where k was the number of iterations, $m_{k+1}$ and $m_{k}$ referred to the momentum terms while $w_{k+1}$ and $w_{k}$ referred to the connections weights in two successive iteration k+1 and k, γ was the momentum factor, η was the learning rate, E was the loss function. The iterative process stopped when the predetermined maximum number of iterations was reached, or the accuracy requirement was satisfied. The values of the weights and bias terms for the NN model were optimized by the training process.

We used the MATLAB software and its neural network toolbox to create our GDM-based neural network model. The model consisted of three layers: an input layer with four nodes, a hidden layer with six nodes, and an output layer with one node. The input variables were connected to the output variable via the neural network. We used a hyperbolic tangent as the transfer function, and the output of a node was updated with the following equations [33,34]:(6)$$X=\overset{n}{\underset{i=1}{\sum }}w_{i}x_{i}+b$$
(7)$$\hat{y}=\tanh\left(X\right)=\frac{e^{X}-e^{-X}}{e^{X}+e^{-X}}$$
where $x_{i}$ was the input of the node, $w_{i}$ was the related connection weight, b was the bias value, n was the number of the nodes in the previous layer, $\hat{y}$ was the output of the node. And mean square error (MSE) was used as the loss function:(8)$$E=\frac{1}{m}\overset{m}{\underset{i=1}{\sum }}(y_{i}-\hat{y_{i}})$$
where $y_{i}$ was the target output, $\hat{y_{i}}$ was the output predicted by the model, and m was the size of the data set. For the model construction, data were selected randomly for training and testing of the NN model at a proportion of 70% and 30%, respectively. The learning rate was set as 0.001, and the momentum factor as 0.9. During the training process, all the weights and bias values were initially randomly assigned. The stop condition was reaching the maximum number of 1000 iterations or satisfying the accuracy requirement of 10^−6^.

## 3. Results and Discussion

### 3.1. Single Track Morphology and Classification

In this research, we studied the effect of laser power and scan speed on the quality of the tracks by analyzing the surface morphologies and characteristics of the printed single tracks. Five types of tracks were observed as depicted in Figure 2. The top views and 3D views of the tracks were at the same enlargement factor, respectively. The outlines showing the fluctuation in laser beam travel direction also used the same vertical scale. Figure 3 is a dot distribution map to show the combinations of laser power and scan speed at which different types of tracks are formed. The experimental figures of all the parameter combinations were demonstrated in Appendix A.

During the SLM process, the energy input increases with laser power and decreases with scan speed. When laser power was low and scan speed was high, experimental results showed that the printed single track of type I was discontinuous and badly bonded to the substrate. The energy input was not sufficient to fully melt the powder layer in the circumstances. As scan speed decreased slightly, the energy input melt through the powder layer but balling phenomenon happened. Balling phenomenon was the manifestation of Plateau-Rayleigh capillary instability in nature’s way [3,4], which occurred when there was a lack of energy input [10] and resulted in bad performance of the fabricated parts. The track of type II seemed to be made of “balls”. As the speed decreased further or the power increased, the track of type III remained discontinuous and became occasionally broken, which was a transition morphology from the track made of “balls” to the continuous one. When the higher laser power was applied, the track of type IV become continuous. The bulged track with an obvious fish-scale pattern was formed. Adequate energy input induced stable melt pool and resulted in tracks with good surface morphology and quality. With the further decrease of scan speed or the further increase of laser power, the track of type V with a fish-scale pattern and a flat top was observed. The large ratio of the width to the height of the track of type V represented that there was an excessive energy input during the fabrication process, which may result in violent fluctuation of melt pool, spatters, rapid loss of elements with a low boiling temperature, finally the reduced performance of the SLM-fabricated part. From the above discussion, the tracks of type IV had the potential to form parts with excellent performance. Any parameter combinations at which the track of type IV was formed were considered to be feasible for the further SLM process.

### 3.2. Analysis of the Evaluation Indicators

For each track, 2048 cross-sections were obtained and data of the width and height of them were extracted. The average and standard deviation values of both the width and height were then calculated and used to obtain the ratios, $R_{w}=s_{w}/\bar{w}$, $R_{h}=s_{h}/\bar{h}$.

Then we confirmed four evaluation indicators $\bar{w}$, $\bar{h}$, $R_{w}$, and $R_{h}$, which were highly-correlated to the surface morphology and key geometrical characteristics of the printed single track. The values of these four indicators for all the parameter combinations were demonstrated in Appendix A in the supplementary file. $\bar{w}$ and $\bar{h}$ represented the dimension of the track, while $R_{w}$ and $R_{h}$ represented the overall fluctuation of the track, which resulted from the fluctuation of energy absorption, non-uniform powder distribution, dynamic fluctuation induced by the recoil pressure and Marangoni effect [6,37], and some other fluctuation during the fabrication.

The results of these indicators were plotted in Figure 4. In the domain of the selected parameters in this experiment, the width increased when the power increased or the speed decreased, revealing a positive relationship between the width and the energy input. However, for a particular laser power, with the decrease of scan speed (the increase of the energy input), the height increased first and then decreased. The dimension of the track was determined by the physical processes during the fabrication, including melting, spreading and solidification. In the practical fabrication, the metal powder melted under the irradiation of the micro-beam laser [38]. The melt volume determined the initial width and height of the melt pool. After the laser beam moved away, spreading and solidification processes simultaneously occurred [39], and the solidification could restrict the spreading process. The spreading process increased the width and decreased the height of the melt pool. After the melt completely solidified, the final dimension of the track was determined. When the energy input increased, the melt volume increased, resulting in the increase of the width and height of the melt pool. At the same time, the spreading was promoted, leading to the increase of the width and the decrease of the height. The decrease of the height caused by melt spreading and the increase brought by the rise of melt volume occurred simultaneously. The turning point of the height in Figure 4b meant the main determinant changed from melt volume to melt spreading. The values of $R_{w}$ and $R_{h}$ matched well with the surface morphologies and characteristics of the tracks, since the continuous tracks of type IV and type V had small values while the discontinuous tracks of type I, type II and type III had large ones.

### 3.3. Model Training and Testing

Among all the ML models, neural network (NN) model was selected in this work. The classification results were used as target output while the four indicators serve as input variables of the NN model. The structure of the model was described in Section 2.4.

We processed a normalization of the data for each indicator by dividing each variable by its maximum value. A dot distribution map showing the normalized results of the four evaluation indicators, which were served as the input variables of the NN model, is displayed in Figure 5. Obviously, the track of type IV which had the potential to form parts with excellent performance possessed a small $R_{w}$ and $R_{h}$, and a relatively large $\bar{w}$ and $\bar{h}$. The classification results of the tracks were used as the target output of the NN model. We used value “1” and “0” to represent the classification results: as tracks of type IV had the potential to form parts with excellent performance, the combinations of laser power and scan speed at which they formed were marked as “1”, while other combinations as “0”, as displayed in Table 1. During the prediction process, the closer that the predicted results from the trained NN model is to 1, the more likely that the parameter to be examined can fabricate track with good surface quality.

After the training process, the NN model was tested by inputting the normalized results of the evaluation indicators of nine parameter combinations of laser power and scan speed. The predicted results from the trained NN model and the corresponding target outputs of the nine parameter combinations were demonstrated in Figure 6a, indicating that they were highly consistent. The percentage of the correct and incorrect predictions were shown in the confusion matrix, as demonstrated in Figure 6b. The column represents the target classification while the row represents the prediction classification. The consistency of the prediction classification and the target classification means the correct predictions of the model. The green squares mean correct predictions and the prediction accuracy of the model was added to be 88.9%, which proved the feasibility of the prediction method. Predicting and decision-making can be realized by using the trained neural network later. After we input the four evaluation indicators of a track fabricated by a certain parameter combination, the ML model will return an output value representing the predicted result of the model. The number serves as the indicator for judging whether this combination can fabricate the track that has the potential to form parts with excellent performance. Expansive works, e.g., increasing the database size, optimizing the existing algorithm, and adjusting the neural network structure, are needed to improve the prediction accuracy.

Taking advantage of the automatic feature-learning capacity of machine learning, this method can obtain a parameter window for single track intelligently and satisfy the demand of the unmanned factory. And the method explored in this paper can be applied to any other materials. It needs to be mentioned that the tracks will interact with the bare plate only at the first layer process during the practical SLM fabrication. In most cases, the tracks will be deposited in the SLM-fabricated layers, which needs further study in the future.

As SLM is a track-by-track and layer-by-layer manufacturing process, the suitable combinations of the laser power and scan speed for a defect-free track is the firm foundation for obtaining a SLM-fabricated part with excellent performance. Multi-track and multi-layer SLM experiments, which take the other process parameters like layer thickness, hatch space and scan strategy into consideration, should be conducted to have a more comprehensive understanding of the key factors that affecting the fabrication quality of the realistic feature in the future. The prediction method developed in this work can also be extended by extracting indicators related to the multi-track and multi-layer SLM experiments and used for the decision-making processes of the other process parameters.

## 4. Conclusions

We proposed a supervised machine learning (ML) method to detect the track defect and predict the printability of material in SLM for the application in the unmanned factory. The developed method helps to figure out the suitable parameter combinations for defect-free printing intelligently. The morphology of the track could be measured by a 3D microscope, which has the potential to be integrated into the selective laser melting system for in-situ measurement. The specific findings are as follows:A prediction method for selective laser melting using machine learning model was developed, which could detect the defect track and predict the printable parameter intelligently.The printed single tracks were classified into five types based on the measured surface morphologies. The classification results were used as target output for the ML model.Four evaluation indicators were determined to evaluate the quality of the tracks quantitatively. They were highly correlated to the surface morphology and key geometrical characteristics of the printed single track.This approach with a backpropagation-based neural network model was successfully used to predict the process parameter window (laser power and scan speed) for TiB2 reinforced AlSi10Mg composite. The feasibility of this prediction method had been proved by experiment.

## Figures and Tables

**Figure 1 materials-13-05063-f001:**
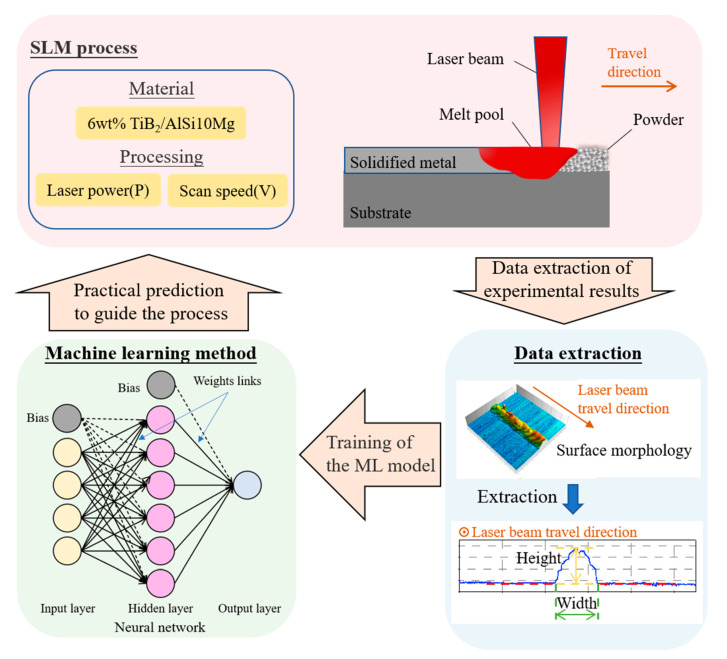
Schematic of the prediction method. The components are the SLM process, data extraction, and machine learning method. After extracted from the experimental results and calculated, the data set is used to train the machine learning model. The trained model can detect the defect track and predict the printability to guide the SLM process.

**Figure 2 materials-13-05063-f002:**
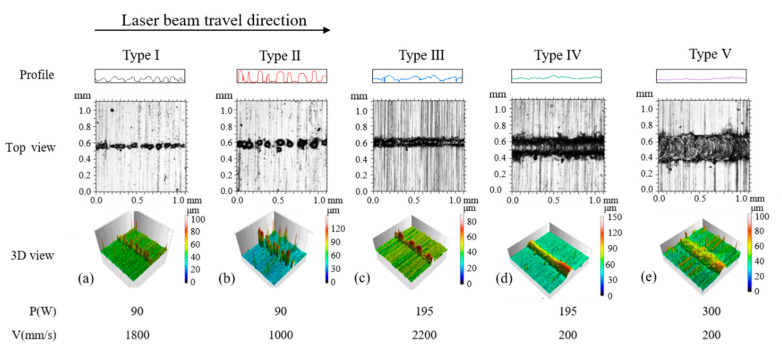
Examples of five types of SLM-fabricated single tracks. Laser power (P) and scan speed (V) were in the range of P = 90–300 W and V = 200–2200 mm/s. (**a**) Type I: discontinuous and badly bonded to the substrate; (**b**) Type II: discontinuous and made of “balls”; (**c**) Type III: discontinuous and occasionally broken; (**d**) Type IV: bulged track with a fish-scale pattern; (**e**) Type V: track with a fish-scale pattern and a flat top.

**Figure 3 materials-13-05063-f003:**
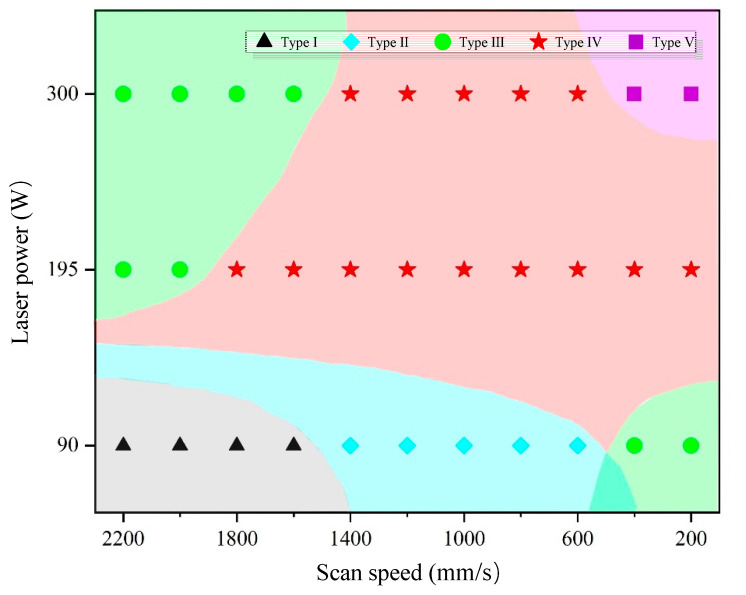
A dot distribution map showing the combinations of laser power and scan speed at which different types of tracks were formed. The base colors are for a clear distinction.

**Figure 4 materials-13-05063-f004:**
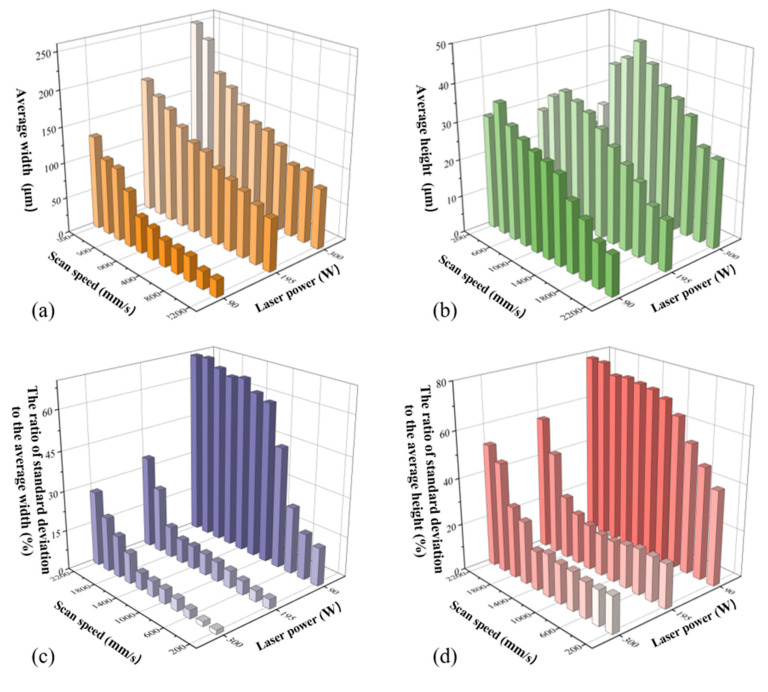
The values of four evaluation indicators: (**a**) the average width, (**b**) the average height, (**c**) the ratio of standard deviation of width to the average width, and (**d**) the ratio of standard deviation of height to the average height.

**Figure 5 materials-13-05063-f005:**
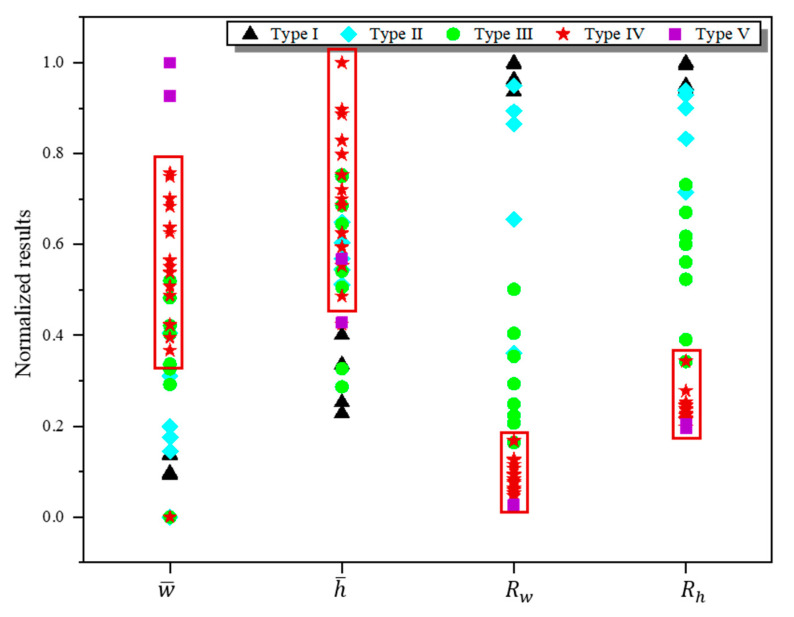
The normalized results of the four evaluation indicators which are served as the input variables of the NN model. The data for each indicator were normalized with dividing each variable by its maximum value. The sets of tracks of type IV were marked with red rectangle outlines. They possessed a small $R_{w}$ and $R_{h}$, and a relatively large $\bar{w}$ and $\bar{h}$.

**Figure 6 materials-13-05063-f006:**
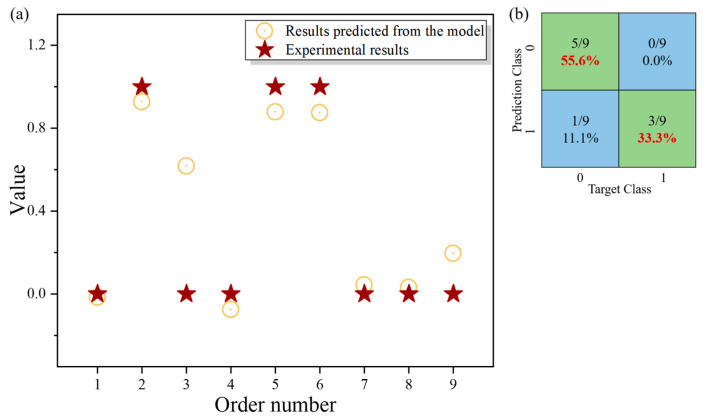
(**a**) The predicted results from the trained NN model and the corresponding target outputs of the nine parameter combinations (the horizontal axis shows the order of the data sets); (**b**) the confusion matrix showing the percentage of the correct and incorrect predictions of the prediction results.

**Table 1 materials-13-05063-t001:** The classification results of the tracks which are used as the target output of the NN model. As tracks of type IV had the potential to form parts with excellent performance, the combinations of laser power and scan speed at which they formed were marked as “1”, while other combinations as “0”.

V (mm/s)		2200	2000	1800	1600	1400	1200	1000	800	600	400	200
**P** **(W)**	300	0	0	0	0	1	1	1	1	1	0	0
	195	0	0	1	1	1	1	1	1	1	1	1
	90	0	0	0	0	0	0	0	0	0	0	0

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
