# Peer review of "Predicting the Printability in Selective Laser Melting with a Supervised Machine Learning Method"

_materials, 2020, doi:10.3390/ma13225063_

Round 1

Reviewer 1 Report

----------------------------------------------------------------------------------------

LINE 106

"Before the printing experiment, pure argon gas was delivered into the chamber until the oxygen content was approximately 0 ppm, aiming to mitigate the oxidation of melt pool during the fabrication."

Comment: How can oxygen content be approximetely 0 ppm. It is either zero, OR approximately some non-zero value. Also, it's hard to believe that oxygen level in any experimental device would be absolute zero. Thus, authors should give some reasonable value here, e.g. 1 ppm, or so..

----------------------------------------------------------------------------------------

* * *

----------------------------------------------------------------------------------------

LINE 233

Z coordinate of Figure 4 d) 3D diagram is wrong. This axis represents the ratio of standard deviation of height to the average height AND not width.

----------------------------------------------------------------------------------------

Author Response

LINE 106

"Before the printing experiment, pure argon gas was delivered into the chamber until the oxygen content was approximately 0 ppm, aiming to mitigate the oxidation of melt pool during the fabrication."

Comment: How can oxygen content be approximately 0 ppm. It is either zero, OR approximately some non-zero value. Also, it's hard to believe that oxygen level in any experimental device would be absolute zero. Thus, authors should give some reasonable value here, e.g. 1 ppm, or so.

Response:

Thanks for the suggestions. In this research, we carried out the experiment with the SLM machine ProX®DMP200 produced by 3D Systems, Inc. The machine is equipped with a built-in oxygen sensor for continuous monitoring of the oxygen content. To mitigate the oxidation content in the melt pool, we delivered pure argon gas into the chamber until the oxygen content detected by the oxygen sensor reached the range between 0 and 1 ppm. Therefore, we used the statement “approximately 0 ppm” in our previous manuscript. The statement has been changed in Section 2.2 in Page 3, listed as follows:

Before the printing experiment, pure argon gas was delivered into the chamber until the oxygen content detected by the built-in oxygen sensor of the SLM machine reached the range between 0 and 1 ppm, aiming to mitigate the oxidation content in melt pool during the fabrication process.

LINE 233

Z coordinate of Figure 4 d) 3D diagram is wrong. This axis represents the ratio of standard deviation of height to the average height AND not width.

Response:

Thanks for the suggestions. The axis of the 3D diagram in the LINE 233 has been corrected. Besides, the authors have thoroughly checked the figures in the manuscript.

Reviewer 2 Report

The authors present a supervised machine learning method to detect the track defect and predict the printability of material in selective laser melting. The manuscript is well and clearly written and the result can have some impact and the laser printing industry.

However, it is not clear why the authors had to classify 5 types of single-track morphologies and how they helped on the machine training. For example, if the best achieved single trach morphology is number IV according to the authors, wouldn't it be enough to account only the other additional type III and V for the machine training?

Also how objective is the SLM quality selection if it is only characterized optically via confocal microscopy?

Figure 6 is difficult to understand, please provide to the readers a more analytic discussion on the text where both value and order number axis are explained.

Author Response

The authors present a supervised machine learning method to detect the track defect and predict the printability of material in selective laser melting. The manuscript is well and clearly written and the result can have some impact and the laser printing industry.

However, it is not clear why the authors had to classify 5 types of single-track morphologies and how they helped on the machine training. For example, if the best achieved single track morphology is number IV according to the authors, wouldn't it be enough to account only the other additional type III and V for the machine training?

Response:

Thanks for the suggestions. During the SLM process, we observed five types of single-track morphologies which could be distinguished from each other. We made a detailed classification of these five types and analyzed the forming process of them for better understanding of the SLM fabrication. The honorable reviewer is right that we can classify all the tracks into three types of single-track morphologies for the machine training. For example, the categories can be tracks fabricated under inadequate, adequate, and excessive energy, respectively. However, for the purpose of a high efficient machine learning model, we considered to divided the tracks into just two categories. Finally, these five types of single-track morphologies were divided into two categories: the best single track (type IV) and the others. And the target output of the model in the combinations of laser power and scan speed at which type IV was formed was marked as ‘1’, while that in the other combinations was marked as ‘0’.

Also how objective is the SLM quality selection if it is only characterized optically via confocal microscopy?

Response:

Thanks for the suggestions. For the tracks of type â…Ł which show a good surface morphology characterized via the confocal microscopy, we have also observed the cross sections of them. Fig. R1 shows two representative examples of these cross sections. As shown, internal defects such as pores are not obvious, which validates that type â…Ł is the best type for high quality SLM printing.

Fig. R1 the cross sections of the single tracks: (a) laser power = 195W, scan speed = 200mm/s; (b) laser power = 195W, scan speed = 400mm/s.

Figure 6 is difficult to understand, please provide to the readers a more analytic discussion on the text where both value and order number axis are explained.

Response:

Thanks for the suggestions. We have provided a more analytic discussion about figure 6 and a detailed introduction about the model building process has been added to Section 3.3 from Page 8 to Page 10, listed as follows:

Among all the ML models, neural network (NN) model was selected in this work. The classification results were used as target output while the four indicators serve as input variables of the NN model. The structure of the model was described in section 2.4.

We processed a normalization of the data for each indicator by dividing each variable by its maximum value. A dot distribution map showing the normalized results of the four evaluation indicators, which were served as the input variables of the NN model, is displayed in Fig. 5. Obviously, the track of type â…Ł which had the potential to form parts with excellent performance possessed a small and , and a relatively large and . The classification results of the tracks were used as the target output of the NN model. We used value '1' and '0' to represent the classification results: as tracks of type â…Ł had the potential to form parts with excellent performance, the combinations of laser power and scan speed at which they formed were marked as ‘1’, while other combinations as ‘0’, as displayed in Table 1. During the prediction process, the closer that the predicted results from the trained NN model is to 1, the more likely that the parameter to be examined can fabricate track with good surface quality.

Figure 5. The normalized results of the four evaluation indicators which are served as the input variables of the NN model. The data for each indicator were normalized with dividing each variable by its maximum value. The sets of tracks of type â…Ł were marked with red rectangle outlines. They possessed a small and , and a relatively large and .

Table 1. The classification results of the tracks which are used as the target output of the NN model. As tracks of type â…Ł had the potential to form parts with excellent performance, the combinations of laser power and scan speed at which they formed were marked as ‘1’, while other combinations as ‘0’.

V(mm/s)

2200

2000

1800

1600

1400

1200

1000

800

600

400

200

P

(W)

300

0

0

0

0

1

1

1

1

1

0

0

195

0

0

1

1

1

1

1

1

1

1

1

90

0

0

0

0

0

0

0

0

0

0

0

After the training process, the NN model was tested by inputting the normalized results of the evaluation indicators of nine parameter combinations of laser power and scan speed. The predicted results from the trained NN model and the corresponding target outputs of the nine parameter combinations were demonstrated in Fig. 6(a), indicating that they were highly consistent. The percentage of the correct and incorrect predictions were shown in the confusion matrix, as demonstrated in Fig. 6(b). The column represents the target classification while the row represents the prediction classification. The consistency of the prediction classification and the target classification means the correct predictions of the model. The green squares mean correct predictions and the prediction accuracy of the model was added to be 88.9%, which proved the feasibility of the prediction method. Predicting and decision-making can be realized by using the trained neural network later. After we input the four evaluation indicators of a track fabricated by a certain parameter combination, the ML model will return an output value representing the predicted result of the model. The number serves as the indicator for judging whether this combination can fabricate the track that has the potential to form parts with excellent performance. Expansive works, e.g., increasing the database size, optimizing the existing algorithm, and adjusting the neural network structure, are needed to improve the prediction accuracy.

Figure 6. (a) The predicted results from the trained NN model and the corresponding target outputs of the nine parameter combinations (the horizontal axis shows the order of the data sets); (b) the confusion matrix showing the percentage of the correct and incorrect predictions of the prediction results.

Reviewer 3 Report

This paper deals with machine learning of SLM process parameters for the single bead deposition process of Al-matrix composite.

Although the paper is in general well written and is containing very interesting results, the reviewer feels that the paper can be improved if the followings are further considered:

  1. This paper focused only on a single bead deposition process. However, considering the SLM building of realistic features, the process parameters which are adequate for the single bead deposition process is somewhat minimum requirement. From the second and the later laser beads in the layer, the bead morphology is influenced significantly by the adjacent laser bead which is deposited in the previous process. It should be also point out that the laser bead will interact with the baseplate material only at the very first layer deposition process, when the multiple SLM layers are deposited. Here in this paper, the baseplate material is different to the deposited material. The reviewer would like to recommend to mention the above limitations in the paper and provide some discussions in the future works.
  2. The authors used very special material, i.e. TiB2 reinforced AlSi10Mg heat resistance aluminum composite, for the material to be deposited in the experiment. The Al 5083 alloy was selected for the base plate. But there is no explanation in the paper about the main reason of selecting those materials. The general properties and possible applications of the selected alloy and composite should be also discussed in the paper. 
  3. The presentations of the data in the paper is generally good.  But in Section 3.3, regarding the Figure 5, the authors presented ‘Normalized results’ without any explanation of how they have normalized them. More explanations are needed.

Author Response

This paper deals with machine learning of SLM process parameters for the single bead deposition process of Al-matrix composite.

Although the paper is in general well written and is containing very interesting results, the reviewer feels that the paper can be improved if the followings are further considered:

1/This paper focused only on a single bead deposition process. However, considering the SLM building of realistic features, the process parameters which are adequate for the single bead deposition process is somewhat minimum requirement. From the second and the later laser beads in the layer, the bead morphology is influenced significantly by the adjacent laser bead which is deposited in the previous process.

Response:

Thanks for the suggestions. The honorable reviewer is right that the process parameters for single tracks is only a part of the adequate parameter set for building the realistic feature. As SLM is a track-by-track and layer by layer manufacturing process, we believe that a defect-free track is the firm foundation of a SLM-fabricated part with excellent performance. And this is why we conduct the research on the single track in this manuscript. The authors are now carrying out the research on multi-track and multi-layer SLM experiments, which take the other process parameters, e.g., layer thickness, hatch space and scan strategy, into consideration. The results will be published in the near future. And we had also added the following introduction to Section 3.3 in Page 10, listed as follows:

As SLM is a track-by-track and layer-by-layer manufacturing process, the suitable combinations of the laser power and scan speed for a defect-free track is the firm foundation for obtaining a SLM-fabricated part with excellent performance. Multi-track and multi-layer SLM experiments, which take the other process parameters like layer thickness, hatch space and scan strategy into consideration, should be conducted to have a more comprehensive understanding of the key factors that affecting the fabrication quality of the realistic feature in the future. The prediction method developed in this work can also be extended by extracting indicators related to the multi-track and multi-layer SLM experiments and used for the decision-making processes of the other process parameters.

It should be also point out that the laser bead will interact with the baseplate material only at the very first layer deposition process, when the multiple SLM layers are deposited. Here in this paper, the baseplate material is different to the deposited material. The reviewer would like to recommend to mention the above limitations in the paper and provide some discussions in the future works.

Response:

Thanks for the suggestions. We had supplemented this limitation to Section 3.3 in Page 10, listed as follows:

It needs to be mentioned that the tracks will interact with the bare plate only at the first layer process during the practical SLM fabrication. In most cases, the tracks will be deposited in the SLM-fabricated layers, which needs further study in the future.

2\The authors used very special material, i.e. TiB2 reinforced AlSi10Mg heat resistance aluminum composite, for the material to be deposited in the experiment. The Al 5083 alloy was selected for the base plate. But there is no explanation in the paper about the main reason of selecting those materials. The general properties and possible applications of the selected alloy and composite should be also discussed in the paper.

Response:

Thanks for the suggestions. As Al-Si alloy has the properties of low density, high plasticity, and relatively high strength, it can satisfy the demand of the aerospace industry, which is an important application field of SLM at present. The TiB2 reinforced Al-Si composite has been proved with the potential to fabricate parts with excellent strength and plasticity in our previous publications[R1,R2]. Therefore, we used TiB2 reinforced AlSi10Mg composite to launch our research in this study. Al 5083 alloy was widely used as the base plate in SLM because of the low cost and high accessibility. And we try to have a clear understanding of the practical interaction process among the laser, powder and base plate when printing the first layer by using the same material.

References are listed as follows:

[R1] Li X P , Ji G , Chen Z , et al. Selective laser melting of nano-TiB2 decorated AlSi10Mg alloy with high fracture strength and ductility[J]. Acta Materialia, 2017, 129(Complete):183-193.

[R2] A Y K X , A Z Y B , B Y W , et al. Effect of nano-TiB 2 particles on the anisotropy in an AlSi10Mg alloy processed by selective laser melting[J]. Journal of Alloys and Compounds, 2019, 798:644-655.

According to our literature research, there isn’t an efficient way to find the process parameters windows which is also important to fabricate parts with excellent performance. Therefore, we propose a supervised machine learning method and try to contribute to this problem. Although we used specific materials in this research, the method explored in the paper can apply to any other materials. We have provided a statement to Section 3.3 in Page 10, listed as follows:

Taking advantage of the automatic feature-learning capacity of machine learning, this method can obtain a parameter window for single track intelligently and satisfy the demand of the unmanned factory. And the method explored in this paper can be applied to any other materials.

3\The presentations of the data in the paper is generally good. But in Section 3.3, regarding the Figure 5, the authors presented ‘Normalized results’ without any explanation of how they have normalized them. More explanations are needed.

Response:

Thanks for the suggestions. We have provided more detailed explanations about the normalization to Section 3.3 in Page 8, listed as follows:

We processed a normalization of the data for each indicator by dividing each variable by its maximum value. A dot distribution map showing the normalized results of the four evaluation indicators, which were served as the input variables of the NN model, is displayed in Fig. 5. Obviously, the track of type â…Ł which had the potential to form parts with excellent performance possessed a small and , and a relatively large and .

Figure 5. The normalized results of the four evaluation indicators which are served as the input variables of the NN model. The data for each indicator were normalized with dividing each variable by its maximum value. The sets of tracks of type â…Ł were marked with red rectangle outlines. They possessed a small and , and a relatively large and .

Round 2

Reviewer 3 Report

Thank you for the improved paper. All the concerns raised in the previous review round have been addressed clearly now. The reviewer doesn't have any more comments on this paper.